# Programmed Death Ligand 1: A Poor Prognostic Marker in Endometrial Carcinoma

**DOI:** 10.3390/diagnostics10060394

**Published:** 2020-06-11

**Authors:** Mianxin Chew, Yin Ping Wong, Norain Karim, Muaatamarulain Mustangin, Nurwardah Alfian, Geok Chin Tan

**Affiliations:** 1Department of Pathology, Faculty of Medicine, Universiti Kebangsaan Malaysia, Jalan Yaacob Latif, Bandar Tun Razak, Kuala Lumpur 56000, Malaysia; mianxinchew@gmail.com (M.C.); ypwong@ppukm.ukm.edu.my (Y.P.W.); amar@ppukm.ukm.edu.my (M.M.); nurwardah@ppukm.ukm.edu.my (N.A.); 2Department of Pathology, Hospital Raja Permaisuri Bainun, Jalan Raja Ashman Shah, Ipoh 30450, Perak Darul Ridzuan, Malaysia; norain33@gmail.com

**Keywords:** endometrial carcinoma, PD-L1, prognostic factor, gynaecology, pathology

## Abstract

Endometrial carcinoma is the only gynaecologic malignancy with a raising incidence and mortality, posing a major health concern worldwide. The upregulation of programmed death ligand 1 (PD-L1) on tumour cells causes T-cell suppression, which impedes antitumour immunity, promotes immune cell evasion and enhances tumour survival. The aim of this study was to evaluate PD-L1 expression in endometrial carcinoma and to correlate it with survival rate. A total of 59 cases of endometrial carcinoma were evaluated. Thirty-two cases of non-neoplastic endometrial tissue were included as control. PD-L1 immunohistochemistry was performed on all cases. PD-L1 expression was evaluated on tumour cells and immune cells. PD-L1 was positive in 62.7% (37/59) and 28.8% (17/59) of immune cells and tumour cells, respectively. PD-L1 expression in immune cells was significantly higher in endometrial carcinoma than in non-neoplastic endometrium (*p* < 0.001). Among the patients with endometrial carcinoma, PD-L1 expression in tumour cells was significantly higher in patients who died (10/15, 66.7%) compared to those who survived (7/44, 15.9%) (*p* < 0.001). It is noteworthy to point out that the expression of PD-L1 in tumour cells was significantly associated with a poor survival. This suggests that immunomodulation using PD-L1 inhibitors may be useful in advanced endometrial carcinoma.

## 1. Introduction

Endometrial carcinoma is a heterogeneous group of tumours derived from endometrial glandular epithelium. It is the most common gynaecological cancer in USA, and was estimated to have 61,880 new cases and 12,160 mortalities in 2019 [1]. In Malaysia, endometrial carcinoma is the third most common gynaecologic cancer and is the seventh most common cancer among woman [2]. It is the only gynaecologic malignancy with a rising incidence and mortality [3,4] and pose a major health concern worldwide.

Endometrial carcinomas are divided into type I and type II that possess different aetiologies, clinical behaviour and outcome [5,6]. Type I endometrial carcinoma includes endometrioid adenocarcinoma and mucinous carcinoma that are associated with unopposed oestrogen exposure, and this group accounts for 70–80% of endometrial carcinoma. Type II endometrial carcinoma consists of papillary serous carcinoma, clear cell carcinoma, undifferentiated carcinoma and carcinosarcoma. The latter group is associated with p53 mutation and has a lesser degree of association with unopposed oestrogen exposure [7].

Most of the patients with endometrial carcinoma are diagnosed and treated at an early stage of disease with favourable outcomes. However, some patients are diagnosed at more advance stage and suffer poor outcome with limited treatment options and a low survival rate [8].

Endometrial carcinoma is routinely managed with surgical resection with or without pelvic radiotherapy and has a 5-year survival rate of 95% if the disease is confined to the uterus. However, if the disease has metastasised, the 5-year survival rate reduces greatly to 17% [3]. They usually have a limited response rate to cytotoxic chemotherapy, targeted agents and hormonal therapy. The triplet regime (paclitaxel, doxoruicin, ciplastin) used in endometrial carcinoma had a disappointing overall response rate of 57% and overall survival rate of 15.3 months [9]. On the other hand, targeted agents such as beracizumab/temsirolimus gave a response rate of <24.5%, while hormonal therapy such as taxane showed a response rate between 18 to 34% [10].

Recently, researchers started to explore a new therapeutic approach which is gaining popularity, namely, immunotherapy with a focus on the tumour microenvironment. One of the potential immunotherapies is the immune checkpoint inhibitor. The immune checkpoint is an inhibitory pathway in the immune system that maintains self-tolerance and minimizes damage during physiological responses to pathogens. Cytotoxic T-lymphocytes-associated protein 4 (CTLA-4) and programmed cell death protein (PD-1) are the most clinically relevant immune checkpoint receptors that are currently being investigated [11].

Programmed death ligand 1 (PD-L1) is a transmembrane protein that modulates T-cell response in normal conditions and is expressed in haematopoietic and inflammatory cells. PD-1 is an inhibitory receptor expressed on T-cells following T-cell activation. In chronic infection and cancer, it is sustained in states of chronic stimulation [12]. The interaction between PD-L1 and PD-1 inhibits T-cell proliferation, cytokine production, and cytolytic activity. This leads to the functional inactivation or exhaustion of T-cells and results in downregulation of T-cell response. In cancer, there is upregulation of PD-L1 expression on tumour cells, which causes T-cell suppression, and this impedes antitumour immunity, promotes immune tumour cell evasion, and enhances tumour survival [13].

PD-L1 overexpression is observed in immune cells of malignant neoplasm, such as melanoma [14], renal cell carcinoma [15] and nonsmall cell lung carcinoma [16], which lead to downregulation of host response against the tumour. Recent immunohistochemical studies also show PD-1 and PD-L1 expression in endometrial carcinoma, ovarian and cervical carcinoma [17,18]. Vanderstraeten et al. discovered that PD-L1 expressions in primary, recurrent and metastatic endometrial carcinoma were 83%, 67% and 100%, respectively [19], while Herzog et al. reported that PD-1 expression was 75% and PD-L1 expression was 25% in endometrial carcinoma [17].

To date, anti-PD-L1 therapies are used in several types of carcinoma, namely, metastatic melanoma, nonsmall lung carcinoma (NSCLC) and renal cell carcinoma (RCC). Meta-analysis by Gandini et al. showed that anti-PD-1/PD-L1 therapy has good clinical response in melanoma and nonsquamous NSCLC, while in squamous NSCLC and RCC, they did not show a significant difference in clinical response based on the PD-L1 status [20]. Interruption of the PDL1/PD-1 interaction represents a potential strategy to prevent tumour-specific T-cell response suppression in the tumour microenvironment. However, there is limited data on PD-L1 expression in endometrial cancer. The aim of this study was to determine the expression of PD-L1 in endometrial cancer and to evaluate the utility of PD-L1 as a prognostic biomarker for endometrial cancer.

## 2. Materials and Methods

### 2.1. Study Design

This was a cross-sectional, descriptive study using archival histopathological materials from the Department of Pathology over a period of 5 years. All hysterectomy specimens diagnosed as endometrial carcinoma were included in this study. Cases with insufficient clinical data or unavailability of paraffin-embedded tissue blocks, cases with equivocal features or indefinite diagnosis, patients who had received neoadjuvant chemotherapy prior to tissue sampling, and cases with synchronous malignancy of the ovary were excluded from the study. Normal endometrium was obtained from hysterectomy for leiomyoma.

Clinical data including age of diagnosis, ethnicity, histological diagnosis, tumour grading, tumour staging, treatment and survival status were obtained from the medical record office and integrated laboratory system. This study was approved by the institution research ethics committee (FF-2018-052; 21 February 2018) and the national medical research and ethics committee (NMRR-17-2909-38987; 26 November 2017).

### 2.2. Histological Examination

All cases were stained with haematoxylin and eosin (HE) and reviewed. For each case, one slide with best representative of the lesion was chosen. The corresponding block was then retrieved for PD-L1 staining. An Olympus microscope BX-41 (Life Science Solution, Selangor, Malaysia) was used to determine the immunohistochemical staining in this study.

### 2.3. Immunohistochemistry (IHC) Staining Method

Rabbit monoclonal (28-8) to PD-L1 (Cat. No. ab205921, abcam, Cambridge, UK) was used at a dilution of 1:500. Normal tonsil tissue was used as positive control tissue (Figure 1). Immunohistochemical staining was performed on tissue sections following manufacturer protocol using EnVisionTM FLEX Mini Kit, High pH (Code No. K8023, Dako, Denmark). Primary antibody was diluted to optimal concentration by antibody diluent, Dako REALTM (Code No. S2022, Dako, Denmark). Washing steps between each reagent were performed using EnVisionTM FLEX Wash Buffer 20× (Code No. K8007, Dako, Denmark) diluted to a 1× working solution with deionized water. The 1× 3,3′-Diaminobenzidine (DAB)-containing substrate working solution was prepared by diluting the 50× concentrated EnVisionTM FLEX DAB+ Chromogen with EnvisionTM FLEX TM Substrate Buffer (Code No. K8023, Dako, Denmark).

Tissue blocks were sectioned approximately 3 µm in thickness and mounted on adhesive glass slide, Platinum Pro White (Product No.: PRO-01, Matsunami, Japan). The slides were left to be air-dried in room temperature overnight. The tissue slides were then incubated on hot plate at 60 °C for 1 h. An initial deparaffinization and pretreatment step was performed in the Decloaking Chamber™ NxGen (Ref. No.: DC2012-220V, Biocare Medical, CA, USA) using the EnVision™ FLEX Target Retrieval Solution, High pH (Code No. DM828, Dako, Denmark) with the conditions of temperature = 110 °C and time = 30 min, followed by cooling at room temperature for 30 min and rinsing with running tap water for 3 min. The slides were subsequently incubated with EnVisionTM FLEX Peroxidase-Blocking Reagent (Code No. DM821, Dako, Denmark) for 5 min, followed by washing steps.

Slides were then incubated with primary antibody for 30 min at room temperature, followed by incubation with EnVisionTM FLEX/HRP (Code No. DM822, Dako, Denmark) for 30 min. Sections were then incubated with 1× DAB-containing substrate working solution for 5 min. The slides were then counterstained with haematoxylin 2 (REF 7231, ThermoScientific, USA) for 5 s. Then, the dehydration step was carried out with increasing alcohol solutions (80%, 90%, 100% and 100%) and 2-times xylene. Finally, the slides were mounted using CoverSealTM-X xylene-based mounting medium (Cat. No.: FX2176, Cancer Diagnostics, USA).

### 2.4. Evaluation of Antibodies Staining

The analysis of immunohistochemical staining was performed by two independent observers (1 consultant pathologist (GCT) and 1 trainee pathologist (MXC)) blinded from the original histologic diagnosis. When there were discordant results, the slides were reviewed together, and a consensus was agreed upon.

PD-L1 expression was evaluated in tumour cells and immune cells. Immune cells are inflammatory cells such as lymphocytes, plasma cells and neutrophils, that participate in the defence mechanism in the body. All PD-L1 stained slides were scored for percentage of positive cells (score 0—negative, <1%—1, 1 to 10%—2, 10 to 30%—3, 30 to 60%—4 and 60 to 100%—5) and intensity of staining (score 0—negative, weak—1, moderate—2 and strong—3), as previously described by Allred et al. [21] The intensity score represented the estimated average staining intensity of positive cell. The percentage and intensity scores were eventually added up to produce a final score that ranges from 2–8. A score of ≥2 is regarded as positive.

### 2.5. Statistical Analysis

Statistical analysis was performed using Statistical Package for Social Science (SPSS for MAC version 21.0, SPSS Inc., Chicago, IL, USA). All demographic data were expressed as mean with standard deviations. Categorical data were expressed as numbers of subjects and percentages. Chi-square test were used to compare PD-L1 expression between non-neoplastic endometrium and endometrium cancer, and to study the associations between PD-L1 expression and clinicopathological data. A value of *p* < 0.05 is considered as statistically significant.

## 3. Results

### 3.1. Demographic Data

There was a total of 59 cases of endometrial carcinoma which were comprised of 51 endometrioid carcinomas, 3 serous carcinomas, 2 clear cell carcinomas, 2 mixed carcinomas and 1 mucinous carcinoma. The mean age of the patients was 53.5 ± 12.0 years. Thirty-two cases of non-neoplastic endometrium obtained from hysterectomy for leiomyoma were also included. The mean age of the non-neoplastic endometrium group was 48.9 ± 8.8 years old (Table 1).

### 3.2. PD-L1 Expression Analysis

#### 3.2.1. Endometrial Carcinoma versus Non-Neoplastic Endometrium

Immunohistochemical staining for PD-L1 was performed on 59 endometrial carcinomas and 32 non-neoplastic endometrial tissues. We found that all non-neoplastic endometrial samples were negative for PD-L1 (Figure 1). In contrast, PD-L1 was expressed in 62.7% of the immune cells (*p* < 0.001) and 28.8% of the tumour cells (*p* = 0.001) (Figure 2, Table 2).

#### 3.2.2. Age (≥60 versus <60 Years)

PD-L1 expression in tumour cells was more frequent in patients over 60 years of age compared to patients with endometrial carcinoma who are younger (43.5% vs. 19.4%, *p* = 0.047). On the other hand, regarding the expression in tumour-infiltrating immune cells, there was no statistically significant difference in PD-L1 expression between these age group (*p* = 0.432) (Table 1).

#### 3.2.3. Ethnic Groups

There is no statistically significant difference in PD-L1 expression between different ethnic groups in both the tumour cells (*p* = 0.432) and tumour-infiltrating immune cells (*p* = 0.847) (Table 1).

#### 3.2.4. Type 1 versus Type 2 Endometrial Carcinoma

The frequency of PD-L1 positivity in the tumour cells was higher in type 2 endometrial carcinoma compared to type 1 endometrial carcinoma. However, the difference was not statistically significant. (42.9% vs. 26.9%, *p* = 0.382) On the contrary, PD-L1 expression in tumour-infiltrating immune cells was slightly higher in type 1 endometrial carcinoma than type 2 (63.5% vs. 57.1%, *p* = 0.746) (Table 1 and Table 3).

#### 3.2.5. Grade and Stage of Tumour

In tumour cells, the frequency of PD-L1 expression was significantly higher in grade 2 and 3 endometrial carcinoma compared to grade 1 tumour (Table 1: *p* = 0.01). There was no statistically significant difference between PD-L1 expression with different stages of endometrial carcinoma (*p* = 0.512).

#### 3.2.6. Survival (Alive versus Died of Disease)

PD-L1 expression was significantly higher in patients who died of disease (10/15, 66.7%) than those who survived (7/44, 15.9%) (*p* = 0.001) (Table 1).

#### 3.2.7. Types of Therapy

Hysterectomy with pelvic and aortic lymphadenectomy were performed in all endometrial carcinomas. In addition to hysterectomy, grade 3 endometrioid carcinoma as well as serous and clear cell carcinoma were treated with adjuvant therapy, either chemotherapy alone or concurrent chemotherapy with radiotherapy. The addition of adjuvant therapy also depends on the presence of risk of recurrence such as myometrial and lymphovascular invasions.

## 4. Discussion

PD-L1 is expressed in different types of tumour cells and in the tumour microenvironment, including infiltrating immune cells. The tumour cells can upregulate PD-L1 expression which leads to inhibition of T-cells function and impedes antitumour immunity, subsequently avoiding immune destruction by the immune system. Previous studies showed that PD-L1 expression by tumour cells correlated with poor prognosis, while PD-L1 expression by tumour-infiltrating immune cells was associated with better overall survival [22,23,24,25,26]. In this study, we also found that PD-L1 expression in tumour cells is associated with poorer survival.

This study showed that PD-L1 was positive in 62.7% of tumour-infiltrating immune cells and 37.3% of the tumour cells in endometrial carcinoma (Table 4). This finding is comparable with other studies, such as that of Mo et al., in which PD-L1 was positive in 60% of immune cells and 17.3% of tumour cells [8]. However, they found that 14.3% of normal endometrium also expressed PD-L1 [8]. On the contrary, Vanderstraeten et al. reported that approximately 80% of both tumour cells of endometrial carcinoma and normal endometrium expressed PD-L1 [19].

All of our non-neoplastic endometrium was negative for PD-L1. This finding differs from the two previous studies by Mo et al. [8] and Vanderstraeten et al. [19], which reported that PD-L1 were positive in approximately 14% and 81%, respectively. Although the immunohistochemistry (IHC) method is predominantly used in the detection of PD-L1 expression, the result is affected by preanalytical and analytical variabilities. The significant difference between the current study and the study by Mo et al., 2016 could be due to the different antibody and IHC protocol used. For example, a rabbit polyclonal antibody with a dilution of 1:400 was used by Mo et al., 2016 while in our study, a rabbit monoclonal to PD-L1 was used at a dilution of 1:500. Therefore, it is justified that Mo et al., 2016 could find more PD-L1 positive cases than in the present study, since the polyclonal antibody has a higher sensitivity when detecting low-quantity proteins, while the monoclonal antibody is more susceptible to changes in antigen conformation due to processing or fixation [28].

In the current study, we have observed that the immunoreactivity for PD-L1 was focal in most of the positive cases. According to Wang et al., 2016, there are multiple factors that affect the immunohistochemical expression of PD-L1 in cancer cells. This includes different types of antibody being used, the difference in cut-off value of PD-L1 staining positivity, and the difference in timing and location of tissue sampling [29]. Therefore, even from the same tumour, the PD-L1 expression may differ, and this is due to focal nature of PD-L1 expression in the tumour [29].

Mo and colleagues found that type 2 endometrial carcinoma had a 100% positive rate of PD-L1 [8]. However, our study showed no significant difference in PD-L1 expression between type 1 and type 2 endometrial carcinoma, in both tumour cells and immune cells. In the grading of endometrial carcinoma, PD-L1 expression in both tumour cells and immune cells were significantly higher in grade 2 and 3 compared to grade 1 tumours. This finding is similar to Mo et al. [8]. Interestingly, in the group of patients with endometrial carcinoma who died of disease, 10 of the 14 patients (71.4%) had positive PD-L1 in tumour cells. This is in contrast to the fact that only 7 of the 55 patients (12.7%) had positive PD-L1 among patients who are alive. This suggests that PD-L1 expression in tumour cells may be used as a predictor of patient survival or as a prognostic marker. It should be mentioned that PD-L1 expression in immune cells does not seem to have a clear value as a prognostic factor and is positive in only about two out of three cases of endometrial carcinoma. Apart from endometrial carcinoma, PD-L1 overexpression has been shown in previous studies to be a poor prognostic factor in several other tumours. In a meta-analysis by Wang et al., PD-L1 overexpression was associated with poorer overall survivals in breast cancer, gastric cancer, urothelial cancer and renal cell carcinoma, while in melanoma, hepatocellular carcinoma, and renal cell carcinoma, PD-L1 overexpression was significantly associated with poorer progression-free survivals [30]. However, there were some discrepancies in the prognostic value of PD-L1 for endometrial carcinoma in previous studies. Zhang et al., 2020 [31] reported that high PD-L1 in TCs was associated with better overall survival, while high PD-L1 in TICs was associated with worse overall survival. In contrast, Engerud et al., 2020 [32] reported that PD-L1 expression in tumour cells was significantly associated with higher grade tumours, with no association with overall survival. The discrepancy might be due to the difference in antibody used, the duration of tissue sample storage, and the method used to assess the antibody expression.

A previous study by Yamazawa et al. showed that young women with endometrial cancer had a better outcome due to a significantly higher percentage of early stage cancer and less myometrial invasion in this age group [33]. This study showed that PD-L1 expression in the tumour cells was more common in patients ≥60 years (43.5%) compared to younger patients <60 years (19.4%).

Kanopiene et al., 2015 reported that 17% of endometrial carcinomas have microsatellite instability (MSI) [34]. Subsequently, another study found no difference between PD-L1 expression in MSI and microsatellite stable (MSS) tumours [32].

PD-1 and PD-L1 expressions within the tumour microenvironment are important in the regulation between activation and tolerance of T-lymphocytes during prolonged antigenic exposure [35]. In a study using animal model, the inhibition of PD-1/PD-L1 resulted in an autoimmune reaction [36]. The use of anti-PD-1/PD-L1 in immune-oncology in tumours was recognised as the “breakthrough of the year” in 2013 [37]. Antibodies targeting PD-1/PD-L1 interactions have been approved by the Food and Drug Administration (FDA) in seven types of malignancies, namely, melanoma, nonsmall cell lung carcinoma, renal cell carcinoma, urothelial carcinoma, colorectal carcinoma, head and neck squamous cell carcinoma and Hodgkin lymphoma [35]. To date, nivolumab, pembrolizumab and cemilimab are the FDA approved anti-PD-1 agents, while atezolizumab, durvalumab and avelumab are examples of PD-L1 inhibitors [38].

## 5. Conclusions

In conclusion, we demonstrated that PD-L1 was expressed in endometrial cancer, but not in normal endometrium. PD-L1 has prognostic value in endometrial carcinoma as it is more commonly expressed in higher grade tumours and is significantly associated with a poorer survival. This study also suggests that immunomodulation such as PD-L1 inhibitors may be useful in the treatment of advanced endometrial cancer.

## Figures and Tables

**Figure 1 diagnostics-10-00394-f001:**
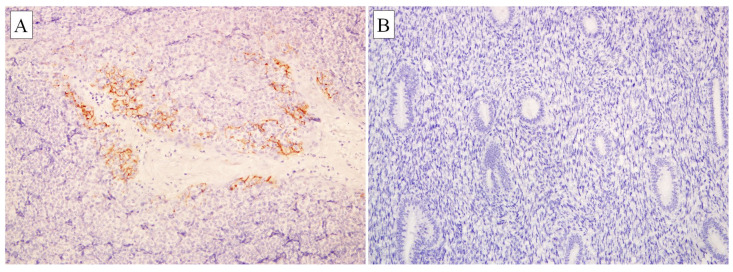
PD-L1 immunohistochemistry showed membranous staining. (**A**) Strong PD-L1 staining in immune cells of tonsils (control; 200×); (**B**) Negative PD-L1 staining in normal endometrium (200×).

**Figure 2 diagnostics-10-00394-f002:**
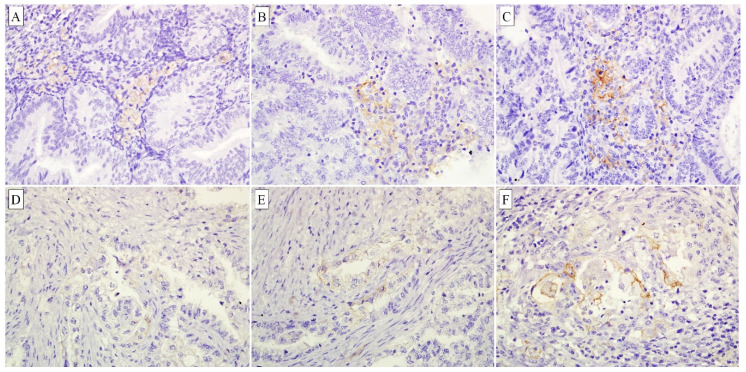
PD-L1 immunohistochemistry showed membranous staining in immune cells and tumour cells. (**A**–**C**) Weak, moderate and strong PD-L1 staining in immune cells (400×); (**D**–**F**) Weak, moderate and strong PD-L1 staining in tumour cells (400×).

**Table 1 diagnostics-10-00394-t001:** Demographic data with programmed death ligand 1 (PD-L1) expression profile in endometrial carcinoma.

Parameters		*n* (%)	PD-L1 Tumour Cells	*p* Value	PD-L1 Immune Cells	*p* Value
Age	Mean ± SD (years)	53.5 ± 12.0				
	<60	36 (61)	7 (19.4)	0.047 *	24 (66.7)	0.432
	≥60	23 (39)	10 (43.5)		13 (56.5)	
Race	Malay	32 (54.2)	10 (31.3)		19 (59.4)	
	Chinese	15 (25.4)	5 (33.3)	0.575	10 (66.7)	0.847
	Indian	12 (20.3)	2 (16.7)		8 (66.7)	
Types of endometrial carcinoma	Endometrioid	51 (86.4)				
	Serous	3 (5.1)				
	Clear cell	2 (3.4)				
	Mixed Mucinous	2 (3.4) 1 (1.7)				
	Type 1 Type 2	52 (88.1) 7 (11.9)	14 (26.9) 3 (42.9)	0.40	33 (63.5) 4 (57.1)	1.0
Grade	1	34 (57.6)	5 (14.7)		18 (52.9)	
	2	9 (15.3)	4 (44.4)	0.01 *	8 (88.9)	0.038 *
	3	16 (27.1)	8 (50)		11 (68.8)	
Stage	1	41 (69.5)	10 (24.4)		25 (61.0)	
	2	7 (11.9)	2 (28.6)	0.57	4 (57.1)	0.512
	3	6 (10.2)	3 (50)		6 (100)	
	4	5 (8.5)	2 (40)		2 (40)	
Survival	Alive	44	7 (15.9)	0.001 *	27 (61.4)	0.767
	DOD	15	10 (66.7)		10 (66.7)	
PD-L1 positive	Immune cells	37 (62.7)				
	Tumour cells	17 (37.3)				

* *p* value < 0.05 is considered as significant; DOD: died of disease.

**Table 2 diagnostics-10-00394-t002:** Expression of PD-L1 in immune cells and tumour cells.

Types	Total Number of Cases (*n*)	PD-L1 Positive in Immune Cells *n* (%)	*p* Value	PD-L1 Positive in Tumour Cells *n* (%)
Endometrial carcinoma	59	37 (62.7)	<0.001 *	17 (28.8)
Non-neoplastic endometrium	32	0 (0)		NA

* *p* value < 0.05 is considered as significant; NA: not applicable.

**Table 3 diagnostics-10-00394-t003:** PD-L1 staining characteristics in tumour cells and immune cells of endometrial carcinoma.

Types of Tumour/Grading	No.	Immune Cells		Tumour Cells		Stage	Survival Status
		Positive (%)	Intensity	Positive (%)	Intensity		
Type I							
Endometroid G1	1	0	-	10	Moderate	1	DOD
	2	0	-	1	Moderate	1	Alive
	3	0	-	0	-	2	Alive
	4	0	-	0	-	1	Alive
	5	0	-	0	-	1	Alive
	6	35	Moderate	0	-	1	Alive
	7	0	-	0	-	1	Alive
	8	10	Strong	<1	Strong	1	Alive
	9	10	Moderate	0	-	1	Alive
	10	<1	Weak	1	Strong	3	Alive
	11	0	-	0	-	1	Alive
	12	0	-	0	-	1	Alive
	13	0	-	0	-	1	Alive
	14	0	-	0	-	1	Alive
	15	1	Strong	0	-	1	Alive
	16	5	Strong	0	-	1	Alive
	17	0	-	0	-	2	Alive
	18	<1	Weak	0	-	1	Alive
	19	10	Strong	0	-	1	Alive
	20	5	Weak	0	-	1	Alive
	21	10	Strong	0	-	1	Alive
	22	1	Strong	1	Strong	1	Alive
	23	<1	Strong	0	-	1	Alive
	24	0	-	0	-	1	Alive
	25	2	Strong	0	-	1	Alive
	26	40	Strong	0	-	2	Alive
	27	0	-	0	-	1	Alive
	28	0	-	0	-	1	Alive
	29	0	-	0	-	1	Alive
	30	<1	Moderate	0	-	2	Alive
	31	2	Strong	0	-	3	Alive
	32	15	Strong	0	-	1	Alive
	33	0	-	0	-	1	Alive
G2	1	30	Strong	10	Strong	4	DOD
	2	1	Moderate	10	Moderate	2	Alive
	3	10	Strong	10	Strong	1	DOD
	4	1	Moderate	10	Moderate	1	DOD
	5	<1	Weak	0	-	2	Alive
	6	0	-	0	-	4	DOD
	7	35	Moderate	0	-	1	Alive
	8	<1	Moderate	0	-	1	Alive
	9	5	Weak	0	-	2	DOD
G3	1	15	Moderate	10	Strong	1	DOD
	2	0	-	<1	Weak	1	Alive
	3	5	Strong	30	Strong	1	DOD
	4	<1	Weak	0	-	3	Alive
	5	<1	Weak	0	-	1	Alive
	6	0	-	<1	Weak	1	Alive
	7	1	Weak	0	-	1	Alive
	8	10	Strong	1	Weak	4	DOD
	9	5	Strong	0	-	1	Alive
Mucinous	1	0	-	0	-	1	Alive
Type II							
Serous	1	<1	Weak	<1	Strong	3	DOD
	2	0	-	0	-	4	DOD
	3	0	-	0	-	4	DOD
Clear cell	1	1	Weak	0	-	1	Alive
	2	15	Strong	<1	Strong	3	DOD
Mixed	1	1	Moderate	0	-	1	DOD
	2	0	-	2	Moderate	2	DOD

No.: Number, G1: Grade 1, G2: Grade 2, G3: Grade 3, DOD: died of disease.

**Table 4 diagnostics-10-00394-t004:** Comparison of PD-L1 expression in present study with previous studies.

References	PD-L1 Expression		
	Tumour-infiltrating Immune Cells	Tumour Cells	Normal Endometrium
Present study (2019)	62.7%	28.8%	0%
Vanderstraeten et al., 2014 [19]	NA	70–80%	81%
Mo et al., 2016 [8]	60%	17.3%	14.3%
Sungu et al., 2019 [27]	36.2%	36.2%	NA

NA: not available.

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
