# Peer review of "Programmed Death Ligand 1: A Poor Prognostic Marker in Endometrial Carcinoma"

_diagnostics, 2020, doi:10.3390/diagnostics10060394_

Round 1
Reviewer 1 Report
The authors described that expression of PD-L1 in tumor cells was significantly associated with a poor survival in endometrial carcinoma. The presentation of this manuscript is very confusing, and many points need to be clarified.
Major points:
- The authors need to provide more information in regard to the type of cancer treatments these patients are receiving in this study. Prognostic marker in endometrial carcinoma must in part influenced by the type of therapy the patients are receiving.
- Important information in regard to tumor microenvironment (TME) such as CD4, CD8, CD68, VEGF as well as PD-L2 are missing.
- The authors use the term “immune cells”. What are these immune cells? What markers were used for the detection of “immune cells”? This needs to be clarified.
- Microsatellite instability (MSI) information is essential information in this type of study and need to be included. It has been shown that MSI was associated with high PD-L1 expression of CD4, CD8, and CD68 cells in TME.
- Many investigators have reported that higher expression of PD-L1 on endometrial carcinoma tissues showed significantly longer PFS. It has also been reported that on 183 primary endometrial carcinomas that high expression of PD-L1 on immune cells was an independent prognostic factor for poor PFS. In addition, a current study shows that high PD-L1 in endometrial carcinomas was associated with better overall survival (OS), whereas high PD-L1 in tumor infiltrating immune cells was associated with worse OS. The authors need to address their finding as compare to these other investigators.
Reviewer 2 Report
In the manuscript “Program death ligand 1: A poor prognostic marker in endometrial carcinoma”, Chew et al. have shown the enhanced level of PDL-1 expression in tumor biopsies compared to normal tissues. Also the expression level of PDL-1 correlates with the higher mortality rate among patients. This is an interesting and short manuscript describing the importance of checkpoint blockade therapy inclusion for these patients.
The manuscript is well written, concised and focus on future interventions using checkpoint blockade immunotherapy. However this is not the very first report on endometrial carcinoma, there have been reports of enhanced checkpoint inhibitor expression in such cancer (doi: 10.1097/PAS.0000000000001395). However I feel as a short communication and given the importance of clinical data, I would accept the manuscript as it hold high significance value of immunotherapy.
- One minor correction : Kindly put first the normal tissue section then followed by biopsies section in the figure. This will be an ideal way to compare the staining.
Author Response
In the manuscript “Program death ligand 1: A poor prognostic marker in endometrial carcinoma”, Chew et al. have shown the enhanced level of PDL-1 expression in tumor biopsies compared to normal tissues. Also the expression level of PDL-1 correlates with the higher mortality rate among patients. This is an interesting and short manuscript describing the importance of checkpoint blockade therapy inclusion for these patients.
The manuscript is well written, concised and focus on future interventions using checkpoint blockade immunotherapy. However this is not the very first report on endometrial carcinoma, there have been reports of enhanced checkpoint inhibitor expression in such cancer (doi: 10.1097/PAS.0000000000001395). However I feel as a short communication and given the importance of clinical data, I would accept the manuscript as it hold high significance value of immunotherapy.
Point 1: One minor correction : Kindly put first the normal tissue section then followed by biopsies section in the figure. This will be an ideal way to compare the staining.
Response:
The figures are arranged in normal tissue first in figure 1, followed by tumour tissue in figure 2. Figure 1 consists 2 normal tissue, i.e. A) normal tonsil and B) normal (non-neoplastic) endometrium, while figure 2 consists of tumour tissue. See Page 5.